

# Comparative transcriptomic analysis of *Lactiplantibacillus plantarum* RS66CD biofilm in high-salt conditions and planktonic cells

Xiaolin Ao[1,*], Jiawei Zhao[1,*], Junling Yan[1], Shuliang Liu[1] and Ke Zhao[2]

[1] College of Food Science, Sichuan Agricultural University, Ya'an, Sichuan, China
[2] Colloge of Resources, Sichuan Agricultural University, Cheng'du', China
[*] These authors contributed equally to this work.

## ABSTRACT

**Background.** *Lactiplantibacillus plantarum* (*L. plantarum*), a dominant strain in traditional fermented foods, is widely used in fermentation industry because of its fast acid production. However, *L. plantarum* is easily inactivated due to acidity, high temperature and other factors. The formation of biofilm by bacteria can effectively increase environmental tolerance. Therefore, it is important to improve the environmental tolerance of *L. plantarum* by studying its biofilm formation conditions and regulatory mechanisms.

**Methods.** After determining a suitable NaCl concentration for promoting biofilm formation, *L. plantarum* was grown with 48 g L$^{-1}$ NaCl. Differential gene expressions in *L. plantarum* biofilm vs. planktonic cells were analyzed using RNA sequencing and validated using qPCR.

**Result.** *L. plantarum* RS66CD biofilm formation formed highest amount of when grown at 48 g L$^{-1}$ NaCl. Altogether 447 genes were up-regulated and 426 genes were down-regulated in the biofilm. KEGG pathway analysis showed that genes coding for D-Alanine metabolism, peptidoglycan biosynthesis, two-component system, carbon metabolism, bacterial secretion system, lysine biosynthesis and fatty acid metabolism were crucial for biofilm formation. In addition, eight other genes related to biofilm formation were differentially expressed. Our results provide insights into the differential gene expression involved in biofilm formation, which can help to reveal gene regulation during *L. plantarum* biofilm formation.

# INTRODUCTION

*Lactiplantibacillus plantarum*, a facultative heterofermentative lactobacilli, is commonly used in fermenting vegetables (*Alan, Topalcengiz & Dğrak, 2018*), cereals (*Oguntoyinbo & Narbad, 2015*), fermented dairy products (*Zhang et al., 2020*). In addition, *L. plantarum* could be linked with some health effects. For example, clinical studies have found that *L. plantarum* CECT7315 and CECT7316 can enhance the immunity of the elderly (*Mane et al., 2011*). The survivability of *L. plantarum* has been found to be influenced by many

Corresponding author
Xiaolin Ao, huavslin@163.com, 245317046@qq.com

factors, e.g., heat treatment, storage temperature and time, acid, microbiota competition and possible presence of bacteriocins or other antimicrobials (*Mona et al., 2020*; *Tripathi & Giri, 2014*). Many microorganisms resist the growth-unfriendly environment by forming biofilms (*Wu et al., 2019*). Therefore, we aimed at improving the environmental tolerance of *L. plantarum* during fermentation and storage by inducing biofilm formation.

Biofilms are composed of microorganisms and an extracellular polymeric matrix that contain polysaccharides, proteins, nucleic acids, and lipids (*Flemming & Wingender, 2010*). Biofilm can effectively increase the stress tolerance of cells. *L. plantarum* JCM 1149 had greater resistance to organic acids, ethanol and sodium hypochlorite inside biofilm than inside planktonic cells (*Kubota et al., 2009*). The biofilm-forming *Lacticaseibacillus rhamnosus* not only showed greater heat resistance, but also maintained higher activity in simulated gastrointestinal tract environment (*Kiew, Cheow & Hadinoto, 2014*). Because the formation of biofilms effectively improves the resistance of microorganisms to extreme environments, studying the mechanism of biofilm formation has become a hot topic.

The formation of biofilms is affected by many factors, such as the roughness of the surface of the carrier, pH, adhesion and gene regulation (*Cappitelli, Polo & Villa, 2014*). Also, inorganic ions affect the formation of biofilms. $Mn^{2+}$ promoted biofilm formation in *L. plantarum* by eliminating oxidative stress and stimulating the secretion of more extracellular polymer substances (EPS) (*Ibusquiza et al., 2015*). The formation of the initial biofilm of *Pseudomonas fluorescens* can be promoted by $Mg^{2+}$, yet higher concentrations lead to blockage of mass transfer channels, affecting the intake of nutrients and inhibiting the further development of biofilm (*Song & Leff, 2003*; *Hoyle, Wong & Costerton, 1992*). $Fe^{3+}$ promoted the secretion of EPS and increased the expression level of genes to improve the stability of biofilms in *Staphylococcus epidermidis* (*Oliveira, France & Ccerca, 2017*). Salt stress by NaCl or KCl promoted biofilm formation in *L. plantarum* and compensated for biofilm phenotypic instability, possibly by regulating the expression of biofilm formation related genes (*Valle et al., 2007*; *Rachid et al., 2000*). In fermenting vegetables, 4%–8% of sodium chloride is commonly applied to improve the flavor and inhibit the growth of spoilage microorganisms (*Xiong et al., 2012*; *Zhang et al., 2016*; *Chun et al., 2020*). Therefore, it is relevant to study the biofilm formation of *L. plantarum* in this concentration range.

In recent years, molecular and post-genomic analyses have reported in detail genes specific to the formation of biofilm. For instance, RNA-Seq analysis of *Enterococcus faecalis* isolates identified 163 differentially regulated genes in biofilm compared with planktonic cells, and that of *Salmonella Typhimurium* showed that 618 genes were over-expressed and 897 genes down-expressed in biofilm compared with planktonic cells (*Seneviratne et al., 2017*). In the biofilm formation of *Bacillus amylobacillus*, the antimicrobial peptide gene *lci* and the drug resistance gene *yvqHI* were among the most up-regulated genes in induction of systemic resistance (*Kröber et al., 2016*). RNA-seq results showed that added tea tree oil caused differential expression of 304 genes in *Staphylococcus aureus* biofilms compared to planktonic cells (*Zhao et al., 2018*). Although biofilms made by pathogens have been studied extensively, analyses of *L. plantarum* biofilm formation are still scarce. Therefore,

we studied the differential genes during the formation of *L. plantarum* RS66CD biofilm at high salt concentration by transcriptomics.

## MATERIAL AND METHODS

### The culture conditions and determination of biofilm formation

*L. plantarum* RS66CD (GenBank MN160322) isolated from traditional Sichuan paocai has good fermentation ability. Previous experimental results showed that *L. plantarum* RS66CD was able to form biofilm (*Zhao et al., 2019*). Therefore, *L. plantarum* RS66CD was selected as the experimental strain. We hope to promote the biofilm formation of *L. plantarum* under the condition of NaCl concentration of 4%–6% (traditional Sichuan paocai fermentation), so as to improve the bacterial salt tolerance. In this experiment, the biofilm formation of *L. plantarum* RS66CD was tested by growing it in modified $MgSO_4$ and $MnSO_4$-free MRS medium with 40, 44, 46, 48, 52, 56 or 60 g $L^{-1}$ NaCl in 96-well cell culture plates at 37 °C. After 24 h incubation, the culture solution was discarded, and biofilms were stained with 50 μL of 0.5% crystal violet for 5 to 10 min, washed with sterile distilled water for 2 to 3 times, after which the plates were dried at 37 °C, and absorbencies were measured at 490 nm.

The concentration of sodium chloride, at which the maximum amount of biofilm was formed, was used in subsequent experiments. Planktonic cells, which were cultured in MRS liquid medium at 37 °C for 24 h (as control treatment) and biofilm formation cells were fixed in 2.5% glutaraldehyde phosphate buffer solution at 4 °C for 8 h, after which water was removed with ethanol. After gold plating, samples were examined using SEM.

### RNA extraction, library preparation, and sequencing

For RNA extraction, cells were collected using centrifugation after culturing at 37 °C for 24 h. Total RNA was extracted using TRIZOL Reagent according the manufacturer's instruction (Invitrogen, California, USA) and genomic DNA was removed using DNase I (TaKara, Shiga, Japan). The quality of RNA determined using 2100 Bioanalyser (Agilent, California, USA). RNA was quantified using the ND-2000 (NanoDrop Technologies).

RNA-seq strand-specific libraries were prepared from 5 μg of total RNA using TruSeq RNA sample preparation Kit from Illumina (San Diego, CA). Shortly, rRNA was removed using RiboZero rRNA removal kit (Epicenter, Stockholm, Swedish) and fragmented using fragmentation buffer. cDNA synthesis, end repair, A-base addition and ligation of the Illumina-indexed adaptors were performed according to the Illumina protocol. Libraries were size selected for cDNA target fragments of 200–300 bp on 2% Low Range Ultra Agarose followed by PCR amplification using DNA polymerase (NEB) for 15 PCR cycles. After libraries were quantified by TBS380, 150 bp*2 paired-end sequencing was done on Illumina Novaseq 6000. The transcriptome sequencing data was submitted to NCBI with the accession number PRJNA626519.

### Quality control and mapping

The raw paired end reads were trimmed and quality controlled using Trimmomatic v.0.36 (http://www.usadellab.org) with sliding window set to length 4 and Phred quality score 15,

and minimum length to 75.The clean reads were separately aligned to the reference genome (*Lactobacillus plantarum* WCFS1, GenBank: AL935263) with orientation mode using Rockhopper software (https://cs.wellesley.edu/~btjaden/Rockhopper/user_guide.html). The expression levels of genes were calculated using Rockhopper with default parameters.

## Differential gene expression analysis

To identify differentially expressed genes (DEGs) in the biofilm vs. planktonic cells, the expression levels of the transcripts were calculated using the fragments per kilobase of read per million mapped reads (RPKM) method. EdgeR (https://bioconductor.org/packages/release/bioc/html/edgeR.html) was used for differential expression analysis. The DEGs were selected using the following criteria: the logarithmic of fold change was greater than 2 and the false discovery rate (FDR) less than 0.05. To understand the functions of the DEGs, GO functional enrichment and KEGG pathway analysis were carried out using Goatools (https://github.com/tanghaibao/GOatools) and KOBAS (http://kobas.cbi.pku.edu.cn/), respectively. DEGs were significantly enriched in GO terms and metabolic pathways when their Bonferroni-corrected *P*-value was less than 0.05. DGEs in biofilm formation of *L. plantarum* RS66CD were further analyzed using hierarchical clustering and a heat map.

## qRT-PCR validation of the transcriptome data

To validate the transcriptome data, six genes were selected for RT-PCR experiments with 16S rRNA gene as the internal reference (Table 1).Total RNA was extracted from *L. plantarum* RS66CD samples using RNA isolation reagent (TIANGEN, Beijing, China) and the first strand cDNA synthesis was carried out using FastKing RT kit (TIANGEN, Beijing, China) according to manufacturer's instructions. RT-PCR amplification using three biological replicates per gene was done in a reaction volume of 20 µL containing 10 µl of YBR Green Supermix, 0.4 µL of forward primer, 0.4 µL of reverse primer, 0.8 µL of cDNA and 8.4 µL of ddH$_2$O. The qPCR conditions were as follows: predenaturation at 95 °C for 10 min, denaturation at 95 °C for 15 s, and annealing and extension at 60 °C for 60 s; fluorescence signals were collected during annealing and extension and the whole process was repeated for 40 cycles. Melting curve analysis is 95 °C for 10 s, 65 °C for 5 s, and 95 °C for 50 s.

## The statistical analyses

All the experiments were conducted in triplicate. Data related to the amount of biofilm formation was tested using Duncan's multiple range test in SPSS 22.0 for Windows. All data were reported as mean ± standard deviation. Differences were taken as statistically significant at $P < 0.05$.

# RESULT

## Formation of *L. plantarum* RS66CD biofilm

*L. plantarum* RS66CD formed more biofilm with 48 g L$^{-1}$ NaCl than with other NaCl concentrations (Fig. 1). Scanning electron microscopy (SEM) revealed that the cells grown in the modified MRS with NaCl were clearly embedded in an extracellular matrix and

**Table 1** Genes and primers in the RT-PCR.

| Name | Primer sequence (5′–3′) | Function |
|---|---|---|
| 16S rRNA | F-TGAGTGAGTGGCGAACTGGTG<br>R-GCCGTGTCTCAGTCCCAATG | KT025937 |
| *groEL* | F-GCAAATCGCTTCTGTATCTTC<br>R-CGCTTCCATCCTTATCATTGTC | Chaperonin GroEL (KJ807061) |
| *ftsH* | F- ACGACTAATGTGAGTGTTGCTGAA<br>R- ATCCTGCGATGATGGTGTGG | ATP-dependent metallopeptidase HflB (MTU86407) |
| *gyrB* | F-AGAAGAGGAAGTTAGAGAAGA<br>R-GCATATCCACTGTTATATTGAAG | DNA topoisomerase (ATP-hydrolyzing) subunit B (MF988195 ) |
| *alr* | F-TAGGATCCATCGAACTCAAACACACCTGCGTC<br>R-CGAAGCTTTGGCAATTTCAGTCGACGAGTATC | Alanine racemase (*Cao et al., 2006*) |
| *LlFabZ1* | F- TAACATATGACTAAAAAATACGC<br>R- CAGAAGCTTCAATGCCACATTGC | Regulation of ACP dehydratase (*Ma et al., 2014*) |
| *LlFabZ2* | F-GAACATATGACTGAAGTAAACATTAATG<br>R- GAAAAGCTTACGCCCTAAGGCAAAAG | ACP dehydrated isomerase (*Ma et al., 2014*) |

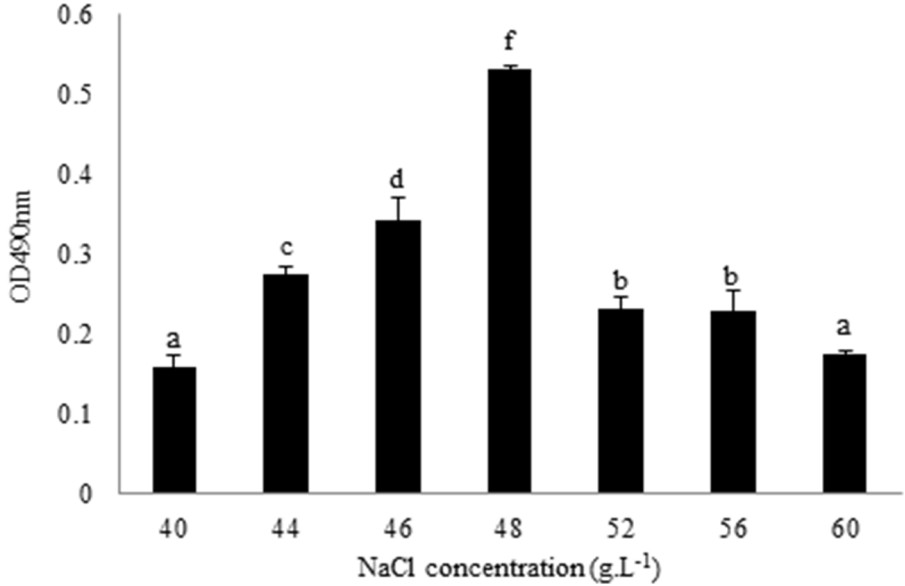

**Figure 1 Quantification of *L. plantarum* RS66CD biofilm at different NaCl concentrations.** The data columns represent means of the three repetitions. The error bars show standard deviation. Different lowercase letters indicate significant difference ($p < 0.05$).

EPS caused the bacteria to aggregate to form microcolonies (Fig. 2), demonstrating that *L. plantarum* could form a distinct biofilm when grown with NaCl.

## Transcriptomic data for biofilm formation of *L. plantarum* RS66CD

On the average, sequencing of the libraries from *L. plantarum* RS66CD biofilm and planktonic cells resulted in $17 \pm 13 * 10^6$ and $9.3 \pm 0.5 * 10^6$ clean reads, respectively, out of

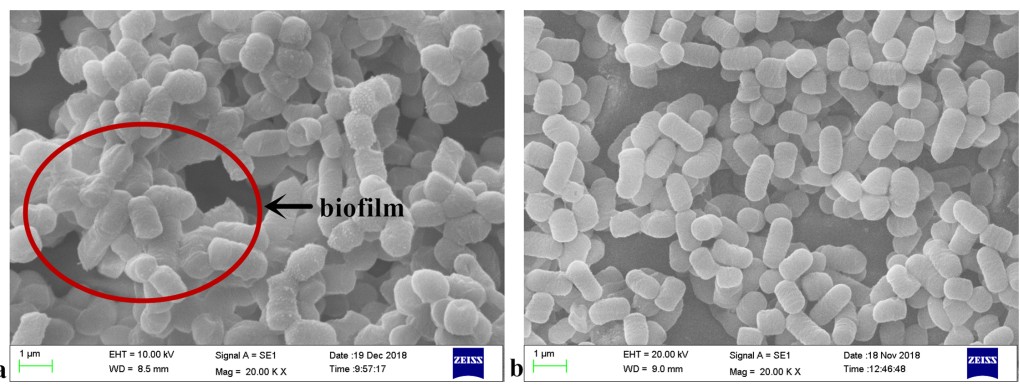

**Figure 2** SEM micrographs of *L. plantarum* RS66CD grown in (A) modified MRS medium with NaCl and (B) MRS medium.

**Table 2** RNA sequencing results for *L. plantarum* RS66CD biofilm and planktonic cells.

| | *L. plantarum* RS66CD biofilm | | | Planktonic cells | | |
|---|---|---|---|---|---|---|
| | **Replicate 1** | **Replicate 2** | **Replicate 3** | **Replicate 1** | **Replicate 2** | **Replicate 3** |
| Clean reads | 8581850 | 10660184 | 32053428 | 9674730 | 9630650 | 8713450 |
| Clean base | 1281768285 | 1591930909 | 4765116422 | 1444240232 | 1436102731 | 1301244000 |
| rRNA (%) | 0.92 | 0.89 | 0.78 | 0.65 | 0.63 | 0.78 |
| Mapped reads | 7038442 | 8769430 | 26643418 | 7805908 | 7837808 | 6956864 |
| Mapped rate(%) | 82.02 | 82.26 | 83.12 | 80.68 | 81.38 | 79.84 |

which 0.86 ± 0.073% and 0.69 ± 0.081% were assigned to rRNA (Table 2). The mapping ratio between sequencing samples and reference genome was above 80%.

## Differential gene expression and KEGG pathway analysis

The transcribed genes were classified into gene ontology (GO) categories: 1047 genes were classified into biological processes, 660 to cellular components and 208 to molecular functions. Compared to the planktonic cells, 447 genes were up-regulated and 426 genes were down-regulated in the biofilm (Fig. 3).

The DEGs were compared with Kyoto Encyclopedia of Genes and Genomes (KEGG, http://www.genome.jp/kegg) and GO databases (http://www.geneontology.org). In the Fig. 4 the abscissa represents the enrichment factor; the vertical coordinate represents the enriched function of the GO term. The larger the circle is, the more differential genes are enriched to this function. Thirty metabolic pathways were enriched in the *L. plantarum* RS66CD biofilm (Fig. 4). In D-alanine metabolism glutathione metabolism, bacterial secretion system, peptidoglycan biosynthesis, two-component system, and carbon metabolism pathways 27 genes were up-regulated and 12 down-regulated in the biofilm compared to the planktonic cells (Table 3, Fig. 5). In addition, five biofilm related genes were up-regulated and four down-regulated in the biofilm (Table 4, Fig. 5).

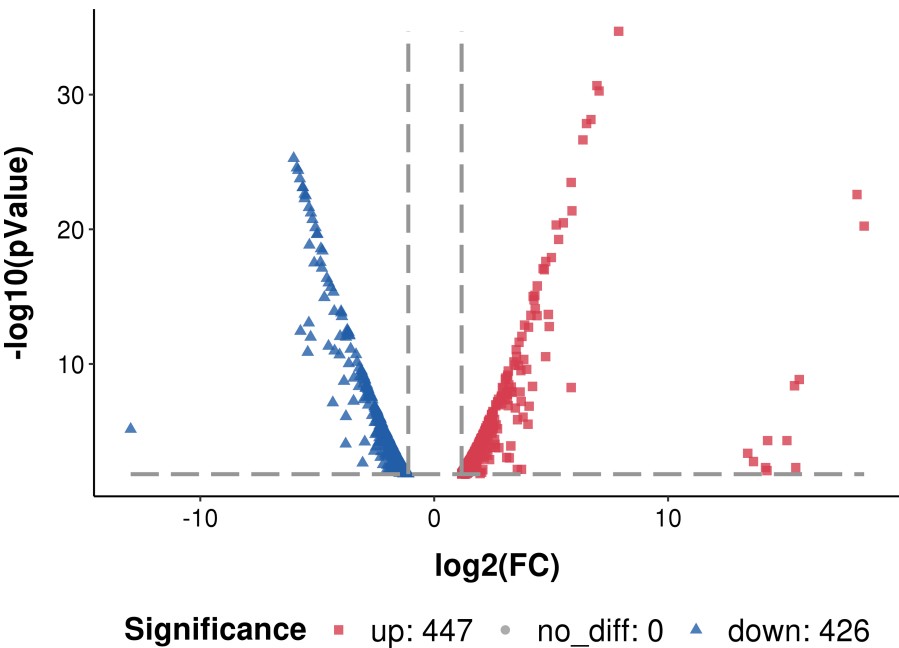

**Figure 3** Differentially expressed genes in *L. plantarum* RS66CD biofilm compared to planktonic cells.
*Log$_2$ FC: fold change in *L. plantarum* RS66CD biofilm compared to planktonic cells.

## qRT-PCR validation of differential expression

Compared to planktonic cells the expression of *groEL* was 2.4 fold and 2.03 times higher in the biofilm as determined by qPCR and RNA-Seq, respectively, the expression of *ftsH* was 1.07 fold and 1.43 fold higher in biofilm, the expression of *gyrB* was 0.96 and 0.83 fold higher, and the expression of *alr* 1.39 and 1.69 fold higher. For *LlFabZ1* and *LlFabZ2* (encode *fabZ* and regulation of ACP dehydratase and ACP dehydrated isomerase) the expression was 1.4 and 4.3 fold higher in biofilm, as determined by qPCR and RNA-Seq, respectively. As the qRT-PCR validation of differential expression was consistent with the RNA-Seq, the transcriptomics results were considered representative.

## DISCUSSIONS

To increase tolerance against adverse environmental conditions, bacteria form biofilms to protect the cells from external forces (*Yan & Bonine, 2019*).The formation of biofilm has several probable mechanisms including hormone induced, environment stimulation, signaling and gene transfer (*Ghafoor, Hay & Rehm, 2011*; *Ramesh et al., 2019*). The main mechanism of biofilm formation can be ascribed to the regulation of transcriptional bacterial physiological metabolism. Biofilms in industrial and clinical context have received considerable attention (*Song, Wu & Xi, 2012*). In this study we characterized the biofilm formation related gene expressionof *L. plantarum* RS66CD.
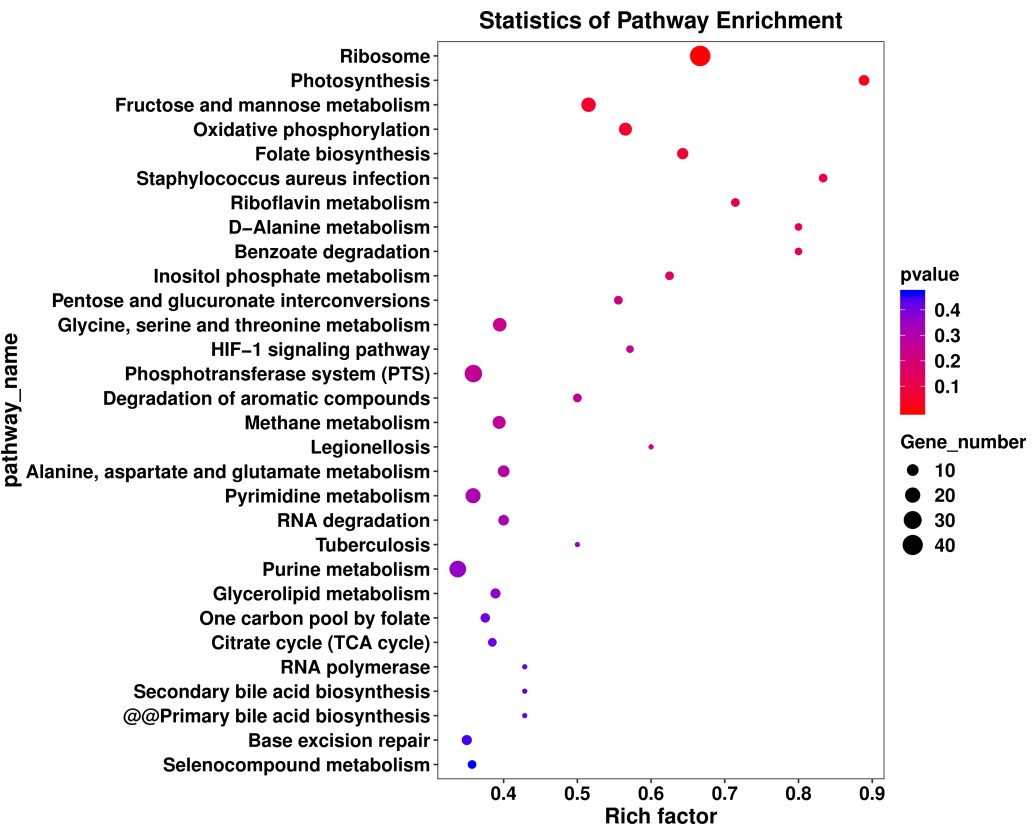

**Figure 4** **KEGG pathway analysis of differentially expressed genes in _L. plantarum_ RS66CD biofilm compared to planktonic cells.** The enriched KEGG categories are on the vertical axis. The enrichment factor, i.e., the ratio of the enriched DEGs in the KEGG category to the total genes in that category, is shown on the horizontal axis.

## D-alanine and peptidoglycan biosynthesis

In our study, D-alanine metabolism and peptidoglycan biosynthesis related genes were up-regulated in the RS66CD biofilm (_dltA_, _dltC_, _ddl_, _alr_). D-alanine, a D-amino acid is abundantly present in the bacterial cell wall (_Laura et al., 2014_). D-alanine is a structural component in bacterial cell wall peptidoglycan. Bacteria modulate surface charge by attaching D-alanine esters to teichoic acids (_Heptinstall, Archibald & Baddiley, 1970_), thus D-alanine affects the formation of biofilm.

Alanylation that increases the positive charge of teichoic acid is mediated by products of the _dlt_ genes (_Nilsson et al., 2016_). In the _dlt_ operon, _dltA_ encodes d-alanine carrier protein ligase, and _dltC_ a d-alanyl carrier proteinthat ensure the ligation of d-alanine to the d-alanyl carrier protein (_Neuhaus & Baddiley, 2004_). Strains lacking the _dlt_ operon have increased antibiotics and acid sensitivity (_Boyd et al., 2000_). Studies have shown that resistance to gentamicin was significantly reduced in the biofilm formed by _Streptococcus mutans_ after the _dltA_ was knockout (_Nilsson et al., 2016_). Interestingly, the reduced antibiotic tolerance of _dltA_ mutants was not due to defects in biofilm formation, but rather to more negative charges on the mutant surface. A future knockout experiment will reveal whether, the
**Table 3 Differentially expressed genes in biofilm formation related KEGG pathways in *L. plantarum* RS66CD biofilm compared to planktonic cells.**

| Term | Gene ID | Name | Log₂FC | Type | Function |
|---|---|---|---|---|---|
| D-Alanine metabolism | | | | | |
| | gene1728 | *dltA* | 1.32 | up | D-alanine ligase |
| | gene2020 | *ddl* | 1.39 | up | D-ala ligase |
| | gene433 | *alr* | 1.69 | up | Alanine racemase |
| | gene1726 | *dltC* | 1.37 | up | D-alanine ligase |
| Peptidoglycan biosynthesis | | | | | |
| | gene422 | *murA* | 1.44 | up | Carboxyvinyl transferase |
| | gene1094 | *dacA* | 2.49 | up | D-alanyl-D-alanine Carboxypeptidase |
| | gene1877 | *mraY* | 1.65 | up | Pentapeptide transferase |
| | gene2020 | *ddl* | 1.39 | up | D-ala ligase N-terminal domain protein |
| | gene707 | *murB* | 1.20 | up | UDP-pyruvate reductase |
| | gene1075 | *lp-1256* | 1.59 | up | Hypothetical protein |
| Two-component system | | | | | |
| | gene508 | *lytR* | 1.76 | up | LytR family transcriptional regulator |
| | gene1358 | *glnA* | −1.70 | down | Type I glutamate-ammonia ligase |
| | gene939 | *citG* | 1.39 | up | Triphosphoribosyl-dephospho-CoA synthase |
| | gene455 | *ftsH* | 1.43 | up | ATP-dependent metallopeptidase |
| | gene3028 | *lamA* | 1.51 | up | DNA-binding response regulator |
| | gene2569 | *lp-3019* | 2.15 | up | Membrane protein |
| | gene2757 | *lrgB* | 1.63 | up | Murein hydrolase |
| | gene2756 | *lrgA* | 2.65 | up | Murein hydrolase |
| | gene3031 | *lamB* | 1.89 | up | Accessory gene regulator AgrB |
| | gene2603 | *hpk* | 2.02 | up | Histidine protein kinase |
| | gene635 | *pstF* | −1.28 | down | Phosphate ABC transporter substrate-binding protein |
| | gene963 | *cydA* | −1.93 | down | Cytochrome D ubiquinol oxidase |
| | gene957 | *mleS* | 4.65 | up | NAD-dependent malic enzyme |
| | gene2658 | *lp-3125* | 3.52 | up | Oxidoreductase |
| | gene964 | *cydB* | −1.79 | down | Cytochrome d ubiquinol oxidase |
| | gene682 | *rpoN* | −1.90 | down | RNA polymerase factor |
| | gene4 | *gyrB* | 0.83 | up | Ica regulatory factor |
| Carbon metabolism | | | | | |
| | gene2989 | *tkt4* | −1.99 | down | Transketolase |
| | gene418 | *sdhB* | 1.44 | up | L-serine dehydratase |
| | gene419 | *sdhA* | 1.31 | up | L-serine dehydratase |
| | gene163 | *serC* | 1.93 | up | Phosphoserine aminotransferase |
| | gene162 | *serA* | 1.80 | up | D-3-phosphoglycerate dehydrogenase |
| | gene1635 | *pps* | −1.58 | down | Phosphoenolpyruvate synthase |
| | gene1840 | *pdhC* | −1.59 | down | Branched-chain $\alpha$-keto acid dehydrogenase |
| | gene1841 | *pdhB* | −1.50 | down | $\alpha$-keto acid dehydrogenase |
| | gene1842 | *pdhA* | −1.19 | down | Pyruvate dehydrogenase |
| | gene1839 | *pdhD* | −1.51 | down | Dihydrolipoyl dehydrogenase |
| | gene1534 | *pgm5* | −1.17 | down | Histidine phosphatase family protein |

**Table 3** (*continued*)

| Term | Gene ID | Name | Log$_2$FC | Type | Function |
|---|---|---|---|---|---|
| Bacterial secretion system | | | | | |
| | gene3155 | *traK* | 2.47 | up | Conjugal transfer protein |
| | gene517 | *radA* | 1.64 | up | DNA repair protein RadA |
| | gene906 | *secY* | 1.35 | up | Preprotein translocase subunit |

**Notes.**
[a]Log$_2$FC: fold change in *L. plantarum* RS66CD biofilm compared to planktonic cells.

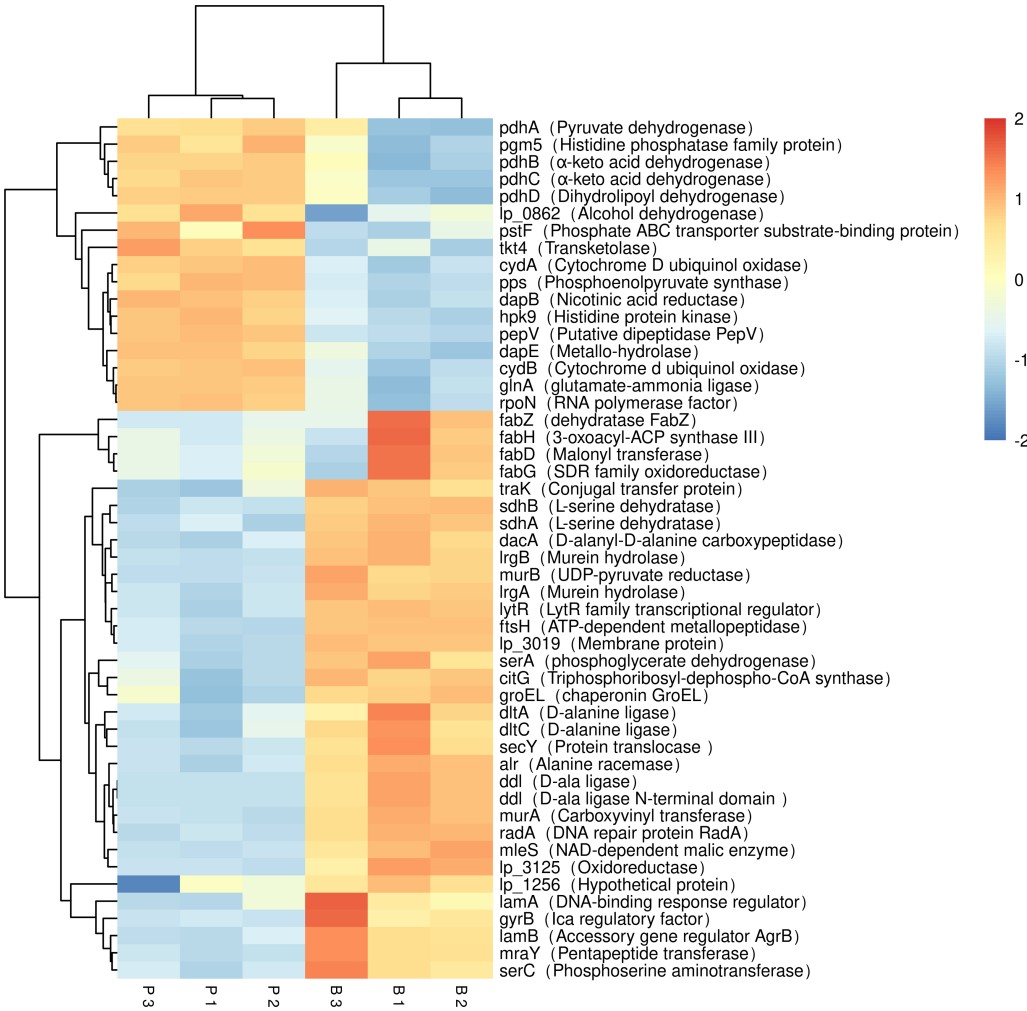

**Figure 5** **Expression of differentially expressed biofilm related genes in *L. plantarum* RS66CD.** Samples in columns and genes on rows. P, planktonic cells; B, biofilm. Differential expression is indicated with the color key.

role of *dltA* of *L. plantarum* RS66CD is similar. Alanine racemase (Alr) converts L-alanine to D-alanine by changing the stereochemical structure of the chiral $\alpha$-carbon atom of L-alanine, and participates in the synthesis of bacterial cell wall peptidoglycan (*Tanner, 2002*). *Ddl* is present in both Gram-negative and -positive bacteria and catalyzes the

**Table 4   Differentially expressed biofilm formation related genes in *L. plantarum* RS66CD biofilm compared to planktonic cells.**

| Gene ID | Name | Log$_2$FC | Type | Function |
|---------|------|-----------|------|----------|
| gene747 | *lp-0862* | −1.32 | down | Alcohol dehydrogenase |
| gene2442 | *dapE* | −1.71 | down | Metallo-hydrolase |
| gene1604 | *dapB* | −1.18 | down | 4-hydroxy-tetrahydrodipicolinate reductase |
| gene1139 | *pepV* | −1.43 | down | Putative dipeptidase PepV |
| gene632 | *groEL* | 2.03 | up | chaperonin GroEL |
| gene1439 | *fabG* | 1.40 | up | SDR family oxidoreductase |
| gene1438 | *fabD* | 1.33 | up | Malonyltransferase |
| gene1435 | *fabZ* | 1.25 | up | dehydratase FabZ |
| gene1436 | *fabH* | 1.33 | up | 3-oxoacyl-ACP synthase III |

**Notes.**

[a]Log$_2$ FC: fold change in *L. plantarum* RS66CD biofilm compared to planktonic cells.

formation of D-Ala-D-Ala dipeptide in an early step in peptidoglycan synthesis, which can promote the synthesis of bacterial cell walls (*Ameryckx et al., 2018*).

The *dacA* encoded D-alanine carboxypeptidase is a low molecular mass penicillin-binding protein (LMM PBP) (*Despreaux & Manning, 1993*). LMM PBPs may regulate cell morphology and mediate the formation of biofilm (*Young, 2006*). *murA* and *murB* in the *mur* enzyme family are involved in the synthesis of propeptides of peptidoglycans(UDP-MurNAc) and the regulation of bacterial osmotic pressure (*Hrast et al., 2014*; *Hattersley et al., 2011*). Our results imply that during the formation of *L. plantarum* RS66CD biofilm, the up-regulation of the peptidoglycan synthesis related genes promote the synthesis of peptidoglycan and improve the stability of the bacteria; plausibly, changes in the cell surface due to the up-regulation of the D-alanine biosynthesis related genes increase cell aggregation and promote the biofilm formation.

## Two-component system

Two-component system is a signaling pathway that regulates the gene expression of bacteria and coordinates responses to environmental stimulus, mainly through adaptation to changes in the environment (osmolarity, light, temperature and oxygen) (*Sieuwerts et al., 2010*), and regulation of developmental pathways and behaviors (sporulation, biofilm formation, competence and chemotaxis) (*Biondi, 2013*).

In our study, the differentially expressed genes related to the two-component system included both up-regulated (*hytR*, *citG*, *lrg* operon, *lam* operon, *ftsH*, *lp-3125*, *mleS*) and down-regulated genes (*pstF*, *cyd* operon, *glnA*, *rpoN*). The gene *citG* is essential for holo-acyl carrier protein (ACP) synthesis in *Escherichia coli* and catalyzes the formation of a prosthetic group precursor from ATP and dephospho-CoA (*Schneider, Dimroth & Bott, 2000*). Bacterial ACPs are the donors of the acyl moiety involved in the biosynthesis of fatty acids, playing an important role in increasing the ability of form biofilm (*Sílvia et al., 2008*). *lrgA* and *lrgB* are presumably involved in the formation of the murein hydrolase transport channel to regulate signal transmission and then to control programmed cell death (*Zhang et al., 2018*). *lamA*, *lamB* and *ftsH* participates in the accessory gene regulator

(AGR) quorum sensing system. The AGR system influences the formation of staphylococcal biofilms during the structuring and dispersion phases (*Cheraghi et al., 2017*). *FtsH* gene may regulate adhesion proteins. Some studies have shown that the *ftsH* gene increases the amount of biofilm formation and is related to multiple environmental stress responses (*Bove et al., 2012*). Further experiments with *ftsH* mutant strain should reveal it affects the formation of biofilm by *L. plantarum* RS66CD and reduces the salt tolerance of the strain. The *agr* operon includes *agrB*, *agrD*, *agrC*, and *agrA*. *agrB* is regarded as a factor to regulate biofilm dispersal, because biofilm biomass is inversely correlated with *agrB* expression (*Li et al., 2018*). *CydA* and *cydB* regulate cytochrome bd ubiquinol oxidase formation. Cytochrome bd panthenol oxidase is a terminal oxidase involved in aerobic respiration of prokaryotes (*Hazan et al., 2016*). It funnels electrons coming from NADH and ubiquinol to cytochrome *c*, and it is also capable of producing significant amounts of the free radical superoxide. *L. plantarum* lacks superoxide dismutase, thus it must reduce the formation of free radical superoxides to promote biofilm formation (*Bazil, 2017*). Due to changes in osmotic pressure, it can also lead to programmed cell death and decreased respiratory intensity. This may explain the down-regulation of *cydA* and *cydB* and the up regulation of *lp_3125* and *mleS* that participate in aerobic respiratory metabolism and simultaneously reduced NAD(P)+ to NAD(P)H. These genes were enriched in this pathway mainly through sense external signal stimulation and influence the expression of quorum sensing system that leads to increases in the adhesion proteins thereby promoting the biofilm formation. This can resist the osmotic pressure changes caused by the addition of NaCl. *L. plantarum* also reduced the production of free radical superoxide by regulating oxidase activity to promote the formation of biofilm and increase the salt tolerance.

## Carbon metabolism

One the roles of carbon metabolism is to help bacteria to adapt to the environment when it changes. In our study, the differentially expressed genes related to the carbon metabolism included both up-regulated and down-regulated genes. The *lytR* family genes that encode putative proteins *FlmA*, *FlmB*, and *FlmC* were up-regulated in the biofilm. The proteins might be involved in cell wall integrity, cell growth, and cell adhesion, and in the developmental process leading to biofilm formation (*Muscariello et al., 2013*). Given the important role of *lytR* family protein genes in the biofilm formation process, the role of *flm* gene must be investigated separately in future experiments. The genes *serA* and *serC* participate in the formation of phosphoserine aminotransferase (P-Ser-HPr) that belongs to the carbohydrate phosphotransferase system. The inactivation of *Neisseria meningitidis hprK* strongly diminished cell adhesion (*Deutscher et al., 2005*). *serA* and *serC* increase the adhesion of strains by increasing the content of P-Ser-HPr, which promotes the formation of biofilm. Pyruvate dehydrogenase(Pdh) participates in many physiological processes, such as carbon metabolism, aerobic respiration and fatty acid metabolism. *PdhA*, *PdhB* and *PdhC* are involved in the formation of a pyruvate dehydrogenase complex which belongs to a family of alpha-keto-acid dehydrogenase complexes (*Neveling et al., 1998*). In our study, the *pdh* genes were down-regulated, which may be related to the change in osmotic pressure inside biofilm, leading to a decrease in respiratory intensity and redox reactions, and to less

intense energy metabolism. Out of the other down-regulated genes *pps* gene catalyzes the phosphorylation of pyruvate and participates in the TCA cycle (*Smyer & Jeter, 1989*). *pgm5* gene regulates histidine phosphatase that is mainly involved in the synthesis of ATP, and *tkt4* encodes a key enzyme of the non-oxidative branch of the pentose phosphate pathway. NADPH participates in reductive biosyntheses and defense against oxidative stress (*Wood, 1968*). These down-regulated genes reduce bacterial damage by improving the oxidative stress response. The biofilm formation can be promoted to resist the change of osmotic pressure by enhancing the bacterial adhesion, participating in the anabolism of serine which constitutes biofilm. Our study found that the genes related to energy metabolism and respiratory intensity were inhibited due to the changes of osmotic pressure.

## Bacterial secretion system

Bacteria secrete many proteins to adapt to their living environment. Bacterial secretion system affects the formation of biofilm by regulating the entry and exit of substances which further affects energy metabolism and signal transmission. Protein export across the cell envelope of Gram-positive (G+) bacteria is relatively simple as only one membrane needs to be passed (*Glöckner et al., 2008*). In our study, three bacterial secretion system genes (*traK*, *radA* and *secY*) were up-regulated in the biofilm. Out of the up-regulated genes, *traK* encodes a transfer protein that regulates substance entry and exit and signaling. The C terminus of *radA* is related to *LonB* protease, an serine protease that regulates capsular polysaccharide synthesis (*Ishibashi et al., 2006*). In many eubacterial *radA* orthologs, active-site serine residue is replaced by alanine, which may be related to the involvement of L-alanine in biofilm formation. *SecY* has been reported to be a transmembrane protein and the most important component of the Sec protein secretion system (*Flower, 2007*).The *secY* encodes a transmembrane protein in the general secretory (Sec) pathway that is responsible for translocation of extracytoplasmic proteins across the plasma membrane in G+ bacteria. The above genes regulate protein secretion to synthesize biofilm, and reduce the damage of osmotic pressure to strain through regulates substance entry and exit. Studies have shown that, the biofilms of *L. plantarum* were reduce the density by destroy extracellular matrix (*Fernández Ramírez et al., 2015*), which further indicates that the formation of related proteins can promote biofilm formation.

## Other key genes

Lysine can inhibit biofilm formation via inducing the production of reactive oxygen species and malondialdehyde (*Ge et al., 2018*). The genes in the Lysine biosynthesis pathway mainly inhibit the synthesis of redox-related enzymes and hydrolases (Table 4). Alcohol dehydrogenase are a group of oxidoreductases which encoded by *lp_0862* gene during the stage of formation (*Hu, Pu & Bai, 2019*).

    Fatty acids are the main components of cell membrane and participate in the physiological metabolism of bacteria. By regulating the type and composition of fatty acids, bacteria can transmit signals, regulate the fluidity of cell membranes, maintain membrane stability, and increase environmental tolerance. In our study, four of the up-regulated genes (*fabG*, *fabD*, *fabH*, *fabZ*) were related to fatty acid metabolism, mainly

involved in the synthesis of type II fatty acids. The genes *fabD* and *fabH* encode ACP transacylase that catalyzes the formation of malonyl ACP, and 3-ketoacyl-ACP synthase III that catalyzes the condensation of malonyl ACP and acetyl-CoA to form 3-keto butyryl ACP, respectively. The enzyme encoded by *fabZ* gene is mainly involved in the cyclic reaction of fatty acid synthesis and promotes a new round of synthetic reactions. The products of fatty acid synthesis include long-chain fatty acids, branched-chain fatty acids and unsaturated fatty acids (UFA). Among them, long-chain fatty acids can be used as signal transduction substances (*Zhou et al., 2015*). UFA has a low melting point which affects the physical properties of cell membranes, making them important molecules in regulating bacterial cell membrane fluidity. Synthesizing branched-chain fatty acids is a way to cope with unfavorable environments (*Santiago et al., 2013*). Comparison of biofilm components of *Staphylococcus aureus* and *Pseudomonas aeruginosaare* with planktonic cells revealed that biofilm growth mode leads to a decrease in the uneven-numbered chain phospholipids, and accumulation of long chain lipids (*Benamara et al., 2011*; *Maria et al., 2019*). This indicates that the accumulation of fatty acids could improve the stability of biofilm. Combined with the results of RNA-Seq, it is shown that *L. plantarum* can affect the fluidity of cell membrane and maintain the stability of biofilm by regulating the type and composition of fatty acids, which can increase the environmental tolerance. Some studies have shown that *fadD* and *fabH* were involved in fatty acid and lipid biosynthesis pathway during the formation of biofilm by *L. plantarum* DB200 (*Angelis et al., 2015*). In addition, the heat stress protein (*GroEL*) also exhibited chloroprene and immune modulatory-properties to improve the aggregation of bacteria and environmental resistance (*Pessione, 2012*). In future experiments, the biofilm formation mechanism can be further analyzed by comparing the fatty acid contents on the surface of planktonic cells and biofilm forming strains.

In summary, this study provided important insights into gene expression during the formation of biofilm by *L. plantarum*. With the formation of biofilm, transcriptome profile changed significantly. Specifically, genes coding for D-alanine metabolism, peptidoglycan biosynthesis, two-component system, carbon metabolism, bacterial secretion system and fatty acid metabolism were found to be crucial for biofilm formation. The results will pave way for a more comprehensive understanding of the mechanisms involved in biofilm formation.

## ACKNOWLEDGEMENTS

The authors acknowledge Associate Professor Petri Penttinen, Sichuan Agricultural University, for his help in revising the manuscript.

### Funding

The authors received no funding for this work.

## Competing Interests

The authors declare there are no competing interests.

## Author Contributions

- Xiaolin Ao conceived and designed the experiments, prepared figures and/or tables, authored or reviewed drafts of the paper, and approved the final draft.
- Jiawei Zhao conceived and designed the experiments, performed the experiments, analyzed the data, prepared figures and/or tables, and approved the final draft.
- Junling Yan and Ke Zhao analyzed the data, authored or reviewed drafts of the paper, and approved the final draft.
- Shuliang Liu conceived and designed the experiments, authored or reviewed drafts of the paper, and approved the final draft.

## Data Availability

The raw measurements are available in Data S1–S3. The transcriptome sequencing data is available at NCBI: PRJNA626519.

## Supplemental Information

Supplemental information for this article can be found online at http://dx.doi.org/10.7717/peerj.9639#supplemental-information.

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

# PeerJ

**Xiong T, Guan QQ, Song SH, Hao MY, Xie MY. 2012.** Dynamic changes of lactic acid bacteria flora during Chinese sauerkraut fermentation. *Food Control* **26(1)**:178–181 DOI 10.1016/j.foodcont.2012.01.027.

**Yan J, Bonine LB. 2019.** Surviving as a community: antibiotic tolerance and persistence in bacterial biofilms. *Cell Host & Microbe* **26**:15–21 DOI 10.1016/j.chom.2019.06.002.

**Young KD. 2006.** The selective value of bacterial shape. *Microbiology and Molecular Biology Review* **70**:660–703 DOI 10.1128/MMBR.00001-06.

**Zhang F, Gao J, Wang B, Huo DX, Wang ZX, Zhang JC, Shao YY. 2018.** Whole-genome sequencing reveals the mechanisms for evolution of streptomycin resistance in *Lactobacillus plantarum*. *Journal of Dairy Science* **101(4)**:2867–2874 DOI 10.3168/jds.2017-13323.

**Zhang QS, Chen G, Shen WX, Wang Y, Zhang WX, Chi YL. 2016.** Microbial safety and sensory quality of instant low-salt Chinese paocai. *Food Control* **59**:575–580 DOI 10.1016/j.foodcont.2015.06.041.

**Zhang SS, Xu ZS, Qin LH, Kong J. 2020.** Low-sugar yogurt making by the co-cultivation of *Lactobacillus plantarum* WCFS1 with yogurt starter cultures. *Journal of Dairy Science* **103**:3045–3054 DOI 10.3168/jds.2019-17347.

**Zhao JW, Ao XL, Cai YM, Liu SL, Chen AJ, Wan H, Xu F, Wang F, He JY. 2019.** Effect of metal ions on the biofilm formation and environmental tolerance of *Lactobacillus Plantarum* RS66CD. *Food and Fermentation Industries* **45(21)**:46–52.

**Zhao XC, Liu ZH, Liu ZJ, Meng RZ. 2018.** Phenotype and RNA-seq based transcriptome profiling of *Staphylococcus aureus* biofilms in response to tea tree oil. *Microbial Pathogenesis* **123**:304–313 DOI 10.1016/j.micpath.2018.07.027.

**Zhou L, Yu YH, Chen XP, Diab A, Wang HH, He YW. 2015.** The multiple DSF-family QS signals are synthesized from carbohydrate and branched-chain amino acids via the FAS elongation cycle. *Scientific Reports* **5**:13294 DOI 10.1038/srep13294.