# Peer review of "Comparative transcriptomic analysis of Lactiplantibacillus plantarum RS66CD biofilm in high-salt conditions and planktonic cells"

_PeerJ, doi:10.7717/peerj.9639_

## Round 0.1 · original submission · Major Revisions

Dear Dr. Ao and colleagues:

Thanks for submitting your manuscript to PeerJ. I have now received three independent reviews of your work, and as you will see, the reviewers raised some concerns about the research. Despite this, these reviewers are optimistic about your work and the potential impact it will have on research studying Lactobacillus and biofilms. Thus, I encourage you to revise your manuscript, accordingly, taking into account all of the concerns raised by all three reviewers.

Please address the very important matter raised by reviewer 2:

“… increasing biofilm formation by a probiotic strain under high osmotic pressure and such high amounts of NaCl can not be considered as a good tool to improve probiotic stability in food products! how can the food with 4-5 percent NaCl be considered healthy?”.

This must be addressed in your revision.

Please avoid using the incorrect references to back up observations or statements. Reviewer 1 provides three examples.

Importantly, please ensure that an English expert has edited your revised manuscript for content and clarity.

Please improve the presentation, clarity and organization of your manuscript (many suggestions raised by the reviewers). Please also work to make the separate subjects flow cohesively. As is, the manuscript is rather disjointed.

Please also ensure that your figures and tables contain all of the information that is necessary to support your findings and observations. Revise incorrect information. The Materials and Methods appear to be missing important information. All statistical methods should be adequately described such that they are repeatable.

I look forward to seeing your revision, and thanks again for submitting your work to PeerJ.

Good luck with your revision,

-joe

·

Basic reporting

The first part of the introduction contains multiple examples of cited articles which are not appropriate for the statement that they are cited for. This undermines the scientific credibility of the statements. Please correct these. Here are three examples:

1) Line 29-31: George et al 2017 is cited to support the claim that L. plantarum is used in many different fermentation processes. The cited article however is a study on sub-inhibitory concentrations of gentamicin for L. plantarum, which is irrelevant for this statement. Please cite either a good review article or a studies that actually proved that L. plantarum is found/used for such a fermentation.

2) Lines 31-34: Two papers are cited after the claim that “L. plantarum plays a vital role in human health” (see also below). However, none of these papers actually discuss the role of L. plantarum in health, they instead focus on other lactobacilli. Lactobacillus is a very broad genus and therefore I am not sure that all these health effects also apply to L. plantarum. Your text does not say this exactly, but by placing these examples behind the claim that L. plantarum plays a vital role in human health, you do seem to insinuate that these health effects apply to L. plantarum as well, which I believe is false. Furthermore the second cited manuscript describes the role of a Lactobacillus EPS in colon cancer and is totally unrelated to the claim that you make regarding “reduce fasting blood glucose levels in diabetic patients”.

3) Shi et al, 2018 is cited to support the claim. The cited article does not mention number of live L. plantarum during production and storage of fermented foods. It instead discusses L. plantarum numbers in a simulated gastrointestinal tract, which is a completely different environment than fermented foods. In addition, I do not believe that this claim is true as many L. plantarum strains start to thrive during the production of fermented foods such as fermented vegetables.

In addition I think that the following claim definitely needs revision:

Line 31: “L. plantarum plays a vital role in human health”. I think this statement is too strong. I believe that L. plantarum could be linked with some health effects, but stating that it plays a vital role seems to be a bridge too far. If you disagree, please provide a good research paper or review that supports this claim.

Finally, the English of the manuscript is of sufficient level to understand the manuscript, but should be proofread a couple of times before resubmission because it does contain several errors that could easily be caught by proofreading:

Lines 13-4: “Lactobacillus plantarum (L. plantarum) that is commonly used producing fermented foods plays a vital role in human health.” should be “Lactobacillus plantarum (L. plantarum), which is commonly used for producing fermented foods, plays a vital role in human health.”
Line 18: “[..] and analysis of differential genes with compared to [...]” should be “[..] and analysis of differential genes compared to [...]”
Line 29: “hetero fermentative” should be “heterofermentative”
Line 31: L. plantarum should be in italic
Line 48: “Mg2+ can promoted the formation of initial biofilm” is grammatically incorrect.
Line 90 -91: Please rewrite this sentence to make it more easy readable.

Apart from this, the goal of the manuscript is clear and seems interesting to pursue.

Experimental design

In my opinion, the methods section fails to meet the standards set by PeerJ. It is missing crucial information which would be necessary for reproduction. Starting with the fact that the sequencing data is not publicly accessible. In addition, the bioinformatic analysis is not thoroughly described, scripts that were used to process the data are missing, tools used are not properly cited and manufacturers names of reagents are often missing. This section needs to be updated and especially the bio-informatic protocols need to be described in more detail.

Here are some specific comments which could help to improve this section:
- The sequencing data is not published in an acceptable discipline-specific repository. In fact, it is completely missing. Please upload your data to ENA and provide an accession number in the manuscript.
- To improve reproducibility please add manufacturers names after each product/platform. E.g. lines 80 -85 miss a lot of information.
- Line 68: GenBank MN160322 points to the 16S rRNA gene sequence. I believe that the whole genome sequence would be better here.
- Line 68: Why was this L. plantarum strain chosen? Because of its probiotic effects or because of its biofilm formation capabilities? Please elaborate
- Line 85: What was the read length used for sequencing?
- Lines 88 -89: What do you mean with “Based on read quality, alignment, saturation, and distribution of reference reads.”? It is unclear what action has been performed on these parameters and I also do not understand the meaning of the word "saturation" in this context.
- Line 92: “The result of the test is in accordance with the difference heterogeneousness criterion (p<0.05 was used to 93 filter statistically significant results)”. I do not understand what you mean with this. Please elaborate.
- Lines 94-95: Please cite the manuscripts coupled with the Goatools and KOBAS tool instead of linking to the software’s homepage.
- Lines 95 - 96: “Biofilm formation of differential expressed genes were used for hierarchical cluster analysis using multiple experiment viewers, and the heat map.” This sentence is unclear to me. I believe that you mean that biofilm formation in the conditions that these genes were differentially express were used for… Please correct.
- Lines 99 - 100: Why was a different method used for RNA extraction and cDNA synthesis for the RT-PCR section? It would make more sense to perform the RT-PCR on the exact same samples used for sequencing. If this was not possible, please explain why.
- Line 107 - 108: “Data was tested using duncan's multiple range test in SPSS 22.0 for Windows”. This is a broad statement that is unclear. What data was tested? And what for?
- Line 109: edgeR software was used to “analyse bioinformatics data”. First, please properly cite this package with the relevant manuscripts. Second, the fact that edgeR was used for differential expression analysis should be mentioned in that section and not here. Third, scripts that explain this analysis should be provided or otherwise please provide a short summary of what steps were performed within the edgeR analysis.
- In general: A complete section is missing which describes how the RNA reads were further processed. To what genome were the reads mapped? How was this genome annotated? How were the mapped reads counted?

Validity of the findings

Due to the shortcomings of the above two sections, I do not feel that I can properly evalute the findings of this manuscript in its current form and have no feedback in the current stage.

·

Basic reporting

In the current manuscript, the authors have investigated the effect of different concentrations of NaCl on biofilm formation by a probiotic L. plantarum strain. The 48 g/L concentration has been then used for further experiments including SEM and the differential gene expression analysis. Finally, qRT-PCR has been performed for validation of transcriptome data which showed a good correlation.

Introduction:
The aim of the study is not reasonable and the problem statement could not vindicate the necessity of the study. Particularly when we see that they have ignored the available data on the topic such as:
De Angelis, M., Siragusa, S., Campanella, D, Di Cagno, R. and Gobbetti, M., 2015. Comparative proteomic analysis of biofilm and planktonic cells of Lactobacillus plantarum DB200. Proteomics, 15(13), pp.2244-2257.
Ramírez, M.D.F., Smid, E.J., Abee, T. and Groot, M.N.N., 2015. Characterisation of biofilms formed by Lactobacillus plantarum WCFS1 and food spoilage isolates. International journal of food microbiology, 207, pp.23-29.
Muscariello, L., Marino, C., Capri, U., Vastano, V., Marasco, R. and Sacco, M., 2013. CcpA and three newly identified proteins are involved in biofilm development in Lactobacillus plantarum. Journal of basic microbiology, 53(1), pp.62-71.

Results:
Line 113 and Fig. 1: What has been the control treatment? The effects of various concentrations of NaCl have been compared to each other. The lack of a suitable control has led authors to select high NaCl concentrations which are not applicable in food industry. Moreover, the letters above the columns should be reordered in Fig. 1.
Line 114 and Fig. 2: The SEM images are not informative and have not made any sense!
Fig. 3. What is log2(FC)? How should we interpret this figure? The figure caption should be written in a way that everybody can understand it without referring to the paper text.
Fig. 4. Please provide an explanation on how to interpret the figure.
Fig. 5. Please add information on the genes investigated to the figure caption. An explanation on how to interpret the figure should be also provided.

Discussion:
The discussion is a literature review rather than making links and comparisons between results of the study and previous works. The reports on the biofilm formation mechanisms in L. plantarum have been completely ignored which is the main weak point of the discussion part.

References:
The references are listed in order of their first mention in the text and the reference list is not alphabetical.
Line 156. Neuhaus and Baddiley et al.: et al should be omitted.
Line 162. Typas et al., 2011: in the reference list the year of publication is 2012.
Line 190. The reference "Bourret and Silversmith, 2010" was not found in the reference list.
Line 354. A.M Stock, et al. was not found in the text.

English syntax:
English throughout the whole manuscript should be revised. Here are some examples:
Line 48. Mg2+ can promoted the…
Line 54. … these result provide a…
Line 60. …under biofilm forming conditions and nisin resistance operon…
Line 62. Treatment…
Line 88. . Based on read quality, alignment, saturation, and distribution of reference reads. (is it a sentence?)
Line 189. HPK on the plasma membrane of the cell to sense external signal stimulation.

Experimental design

The manuscript fits within the journal scope. however, considering the available reports on the mechanisms involved in biofilm formation by L. plantarum especially that of Angelis et al. (2015) entitled "Comparative proteomic analysis of biofilm and planktonic cells of Lactobacillus plantarum DB200", the only novelty of this ms seems to be the in detail investigation of NaCl effect on formation and development of biofilm by this species. However, some questions arise here:

Biofilm formation is important in relation to food spoilage and pathogenic organisms. Although some L. plantarum strains contributing to food deterioration, the strain examined in this study is a probiotic one and the authors intended to induce biofilm formation in this bacterium as a tool to overcome the food-related stresses. But are we allowed to enrich high-salt foods with probiotics? According to CODEX, it is not recommended for foods containing more than 450 mg sodium per serving or 100 g. The recommended daily intake of salt is 6 g/d (=2.4 g/d sodium). The in vitro concentration of 40 or 48 g/L is too high to be used in healthy foods and the alternative methods such as encapsulation seem to be more efficient and reasonable!

Materials and Methods:
Line 73. The MRS medium has been used as a control of Mg and Mn free MRS with added NaCl. But we know that even trace amounts of Mg and Mn pose effects on biofilm formation and the growth ability of bacteria. To exclude their effects, why have not the authors used the MRS medium with added NaCl as a treatment medium?
Line 86. More details should be provided on the differential gene expression analysis.
Table 1. Please provide the target genes and references for each primer set.

Validity of the findings

The main aim of this study is questionable, as increasing biofilm formation by a probiotic strain under high osmotic pressure and such high amounts of NaCl can not be considered as a good tool to improve probiotic stability in food products! how can the food with 4-5 percent NaCl be considered healthy?

Additional comments

no comment

Reviewer 3 ·

Basic reporting

Minor comments:
1. The English language could be somewhat improved to ensure that an international audience can clearly understand the text.
Some examples where the grammar could be improved include lines in the Abstract:
- “analysis of differential genes with compared to planktonic cells by RNA-seq technology (Illumina sequencing)”
- “Lactobacillus plantarum (L. plantarum) that is commonly used producing fermented foods plays a vital role in human health.”
- It is also not clear what you mean with “activity” in the sentence “The biofilm formed by the bacteria can effectively increase the activity of the cells”? Do you mean transcriptional activity?
Line 180: “acids, which play a central role in increasing the ability of to form biofilms (Sílvia et al., 2008).”
Line 257: “This study has no supported by funding sources.”
2. EPS abbreviation should be explained.
3. Please provide more explanation in the figure legends. For example:
Figure 1: Please indicate if the data columns represent means or medians of the three repetitions. For the letters above the bars: these are statistically significant differences, but compared to what (which column is compared to which other column)?
Figure 5: Please explain in the legend what P1, P2, P3 and B1, B2, B3 mean.
4. Figure 2: Please indicate where the extracellular matrix is located in the photo.

Experimental design

Minor comment:
1. In the section “2.2 RNA extraction, library preparation, and sequencing”, manufacturers of the reagents and kits should be mentioned.

Validity of the findings

Major comments:
1. Biofilm formation was promoted by the addition of 48 g.L-1 NaCl. It should be discussed how relevant this is in real life, for example in the industrial setting. Is 48 g.L-1 NaCl a relevant concentration?
2. Biofilm was grown in 48 g.L-1 NaCl in modified, MgSO4 and MnSO4-free MRS medium, while the control condition was grown in MRS. This might be a limitation of the study. Is the transcriptome change due to the high levels of NaCl and absence of MgSO4 and MnSO4, or due to biofilm formation?
3. In the manuscript there is a statement: “KEGG pathway analysis showed that genes coding for D-Alanine metabolism, peptidoglycan biosynthesis, two-component system, carbon metabolism, bacterial secretion system, lysine biosynthesis and fatty acid metabolism were crucial for biofilm formation.” However, changes in gene expression during biofilm formation do not necessarily indicate that these genes are crucial for the biofilm formation itself. These changes might be signs of metabolic changes happening in parallel with the biofilm formation, due to high NaCl concentration, etc.
A more critical view of the findings and additional discussion is encouraged. For example, it should be mentioned what additional in vitro experiments should be performed to validate the findings (e.g. making L. plantarum knock-out mutants in the described genes).

Additional comments

The article poses a relevant research question, as less is known about biofilm formation by beneficial lactobacilli compared to pathogens. An advanced transcriptomic approach is used to reveal the gene expression changes linked to L. plantarum biofilm formation. A wide array of potential hits is identified, and a detailed overview is provided of the genes and their function in the discussion section. This article provides novel data of scientific interest and can pave the way for future studies.

---

## Round 0.2 · Minor Revisions

Dear Dr. Ao and colleagues:

Thanks for resubmitting your revised manuscript. As you can see from the re-reviews, there are a few more issues that need attention.

I agree with many of the concerns of the reviewers, and thus feel that their suggestions should be adequately addressed before moving forward with accepting your work for publication.

I look forward to seeing your revision, and thanks again for submitting your work to PeerJ.

Good luck with your revision,

-joe

·

Basic reporting

- During the revisions of this manuscript, a new nomenclature for Lactobacillus has been published (see https://doi.org/10.1099/ijsem.0.004107 or http://lactobacillus.uantwerpen.be). I strongly encourage the authors to update their manuscript according to this new nomenclature. In this case, Lactobacillus plantarum, would need to be changed to Lactiplantibacillus plantarum.

- Line 46: Lactobacillus rhamnosus is mentioned for the first time so should not be abbreviated. Also according to the newest nomenclature this species is now called Lacticaseibacillus rhamnosus

- Lines 50-60: Several biofilm-promoting agents are presented here, but it is unclear whether these are all related to L. plantarum. Please add the organism names next to each claim for clarity. E.g. Mn2+ promoted biofilm formation in L. plantarum….

-Line 123: “DGEs” should be corrected to “DEGs”.

-Line 241: These species should first be written in full format instead of abbreviated. Also please update to new taxonomy.

Experimental design

- Lines 78-79: Are these experiments published? If so, please add a reference, if not maybe add a statement between brackets (unpublished results).

- Line 108: The data was submitted to NCBI but lacks good medata, which makes it hard for others to reproduce. I believe more effort should be put in providing decent metadata. For example the abstract reads “The RNA-seq for L. plantarum biofilm and planktonic cells”. Providing the abstract from this manuscript would already give better context. In addition, each sample was submitted with another strain isolation name (e.g ”a1”, “a2”, …) which is not correct as only strain RS66CD was used for the experiments, please correct. Furthermore the “sample type” is incorrect. You state “single-cell” while it should be “cell culture” (see https://submit.ncbi.nlm.nih.gov/biosample/template/?package=Microbe.1.0&action=definition). The same goes for the sample description, wherein it would be useful to provide the Na+ concentrations.

- Line 110 - 118: Citing bio-informatic tools is of importance for the developers to be able to keep attracting funding. Please cite the resources you use. There are missing citations for Trimmomatic, Rockhopper, EdgeR, GOAtools and KOBAS.

- Line 159 - 160: Please cite the KEGG and GO databases.

- Line 164: What twelve pathways are putatively involved in the formation of biofilm?

Validity of the findings

- Line 167: figure 5 should be updated. It is hard to interpret the short gene names only and matching them to Table3. Also some genes shown on the figure are not in table 3 (FabZ). The readability of this figure would be improved if the full gene annotation is shown next to the short gene names.

- Lines 170-177: It is unclear to me what the motivation for selecting these specific gene is. Please provide a reasoning on why these genes were specifically targeted with qPCR.

-Lines 178-...: The whole discussion is hard to follow. It reads more like a literature study than a discussion of the research results and is hard to interpret. Please update this section by guiding the reader through your results. For example it is hard to distinguish which genes were upregulated in this study and which were found in literature. You could start each by more specifically pointing out which genes were upregulated and then start discussing them.

-Line 207: Why is dacA relevant here? I do not see it in Figure 5. Is it also a DEG?

Additional comments

I would like to thank the authors for revising their manuscript. I have reevaluated the work and now also reviewed the results and discussion sections, which I did not assess previously. The main weakness of the manuscript in its current state is the discussion. It is really hard to follow and sometimes seems unrelated to the results of the manuscript. Although all genes discussed there are DEGs, it is sometimes unclear how they are related to biofilm formation. Maybe a reduction of this section to only keep the relevant information would be useful.

·

Basic reporting

The revised manuscripts has been well structured. In the revised version, the aim of the study has been well explained. Now, we know that the test strain is not a probiotic one, but still it can be used as a fermentative strain in high-salt fermented foods like the traditional one mentioned in this manuscript. Relevant references have been added to the revised version.
The English syntax has been improved; but there are minor errors in the revised version:
-formation biofilm or biofim formation?
- lines 80-81: Previous experimental results show that L. plantarum RS66CD was able to form biofilm

Experimental design

The revised methods have been expalined in details.

Validity of the findings

The revised version has provided explanations on how to interpret the figures and the questions have been well explained.

Reviewer 3 ·

Basic reporting

The authors made significant efforts to improve the manuscript regarding methods and discussion, as well as language and appropriate references. A few aspects remain:

1. The parts added in the discussion as tracked changes are relevant, however they would benefit from another proofreading, as it is not always clear what the authors mean (e.g. “The biofilms of L. plantarum were reduce the density by destroy extracellular matrix (Fernández Ramírez, et al., 2015), which further indicates that the formation of related proteins can promote biofilm formation”).

2. Figure 1 legend is now better explained, however the meaning of a, b, c... is still not clear. If there is a letter “a” above the column, the result is significantly different compared to what? Please define what each letter means.

Experimental design

No comment

Validity of the findings

3. I am still not entirely convinced by the authors’ reply in the rebuttal to my comment “These changes might be signs of metabolic changes happening in parallel with the biofilm formation, due to high NaCl concentration, etc.”.
The authors should emphasize in their manuscript that this work is focused on gene expression during the formation of biofilm specifically as a result of high salt concentrations, and biofilm formation without such high salt concentrations might involve different mechanisms. It might be appropriate to change the title from “Comparative transcriptomic analysis of Lactobacillus plantarum RS66CD biofilm and planktonic cells” to “Comparative transcriptomic analysis of Lactobacillus plantarum RS66CD biofilm in high-salt conditions and planktonic cells”.

---

## Round 0.3 · accepted · Accept

Dear Dr. Ao and colleagues:

Thanks for revising your manuscript based on the concerns raised by the reviewers. I now believe that your manuscript is suitable for publication. Congratulations! I look forward to seeing this work in print, and I anticipate it being an important resource for groups studying Lactobacillus and biofilms. Thanks again for choosing PeerJ to publish such important work.

Best,

-joe